# Analysis of Oral Health Literacy in Caregivers of Special Needs Individuals in Special Schools and Social Institutions in Jakarta

**DOI:** 10.3390/dj11090221

**Published:** 2023-09-19

**Authors:** Esther Rotiur Hutagalung, Anandina Irmagita Soegyanto, Mas Suryalis Ahmad, Masita Mandasari

**Affiliations:** 1Oral Medicine Residency Program, Faculty of Dentistry, Universitas Indonesia, Jakarta 10430, Indonesia; terrehutagalung@gmail.com; 2Department of Oral Medicine, Faculty of Dentistry, Universitas Indonesia, Jakarta 10430, Indonesia; anandina.irmagita74@ui.ac.id; 3Special Needs Dentistry Program, Universiti Teknologi MARA, Shah Alam 40450, Malaysia; drsuryalis@uitm.edu.my

**Keywords:** caregivers, health status, individuals with special needs, oral health literacy, special care dentistry

## Abstract

Background: Individuals with special needs (IWSN) are susceptible to oral conditions such as caries and periodontal disease. In order to improve oral health of IWSN, it is important to improve the oral health literacy (OHL) of caregivers, as they play an important role in the daily hygiene and personal care of these people. Objective: This study aimed to analyze the OHL in caregivers of IWSN in special schools (informal caregivers) and social institutions for people with disabilities (professional caregivers) in Jakarta, Indonesia. Methods: The study was conducted with a cross-sectional and descriptive analytic design with a cluster sampling method of 400 informal and professional caregivers. The study utilized the validated Health Literacy Dentistry-Indonesian Version (HeLD-ID) questionnaire to measure OHL. Quantitative data was analyzed using non-parametric Kruskal Wallis and Mann Whitney tests (significant level *p* < 0.05). Results: The median total OHL score of respondents was 3.14 (0.24–4) for informal caregivers and 3.21 (0–4) for professional caregivers. The OHL score of the two populations showed significant differences in the domains of receptivity (*p* = 0.036), understanding (*p* = 0.030), and economic barriers (*p* = 0.022). Significant differences in OHL scores were also noted among caregivers according to their sociodemographic characteristics, such as level of education, and number of IWSN handled. Conclusion: Informal and professional caregivers in this study showed good level of OHL. To elucidate the relationship between caregiver’s level of OHL with IWSN, further study is necessary.

## 1. Introduction

Oral health is an important component that integrates with general health, well- being, and quality of life [1,2]. The need for oral health services is also experienced by individuals with special needs (IWSN) who are more prone to oral diseases such as caries and periodontitis [3,4,5]. Generally, IWSN do not make their own decisions and depend on caregivers to assist and monitor daily activities, including general and oral health care due to their limitations [6]. Caregivers play an important role in the oral health of IWSN [2,7,8]. Good oral health knowledge in caregivers will increase their positive attitudes and foster good behavior in maintaining the oral health of the IWSN in their care [6].

Oral health literacy (OHL) is defined as the ability to obtain, process, and understand basic oral health information and services needed to make informed oral health decisions [9,10,11]. In previous literature, it was found that those with low OHL demonstrated poor oral conditions and experienced barriers in accessing healthcare services [12,13,14,15]. Meanwhile, a study by Blizniuk et al. reported that adults with adequate OHL were more likely to complete dental treatment, and thus have better oral conditions [16].

For IWSN, OHL of caregivers is important to ensure maintenance of oral health. This is because many IWSN depend on their caregivers for activities of daily living, including their oral hygiene [17]. Caregivers of IWSN include parents or family members (regarded as informal caregivers), as well as professionals or workers in social or educational institutions (regarded as professional caregivers) [18,19]. It was found that OHL of both informal and professional caregivers of IWSN has a significant impact on oral health of the person under their care [20].

While many studies have reported the level of OHL among caregivers of IWSN, there is a lack of such research in the Southeast Asian region. Furthermore, comparisons in the level of OHL among caregivers of IWSN attending the different types of institutions are also not thoroughly explored. Such investigations are important to ensure that all IWSNs, regardless of institutions that they attend, received adequate care from their caregivers.

This study aimed to investigate and compare the OHL of caregivers of IWSN in special schools and social institutions for people with disability in Jakarta, Indonesia. Findings of this study would provide important information for policy makers, governing agencies and authoritative bodies to identify areas for development, so that the oral health of IWSN can be further improved and maintained through active and effective involvement of caregivers.

## 2. Respondents and Methods

### 2.1. Study Design

This was a cross-sectional and descriptive analytic study, using a paper-based validated questionnaire, involving caregivers of IWSN population in 14 special schools and 11 social institutions for people with disability in Jakarta, Indonesia. Ethical clearance was obtained from the Faculty of Dentistry Universitas Indonesia Ethical Committee (No. 40/Ethical Approval/FKGUI/IX/2021).

### 2.2. Respondents

This study involved parents or family members (informal caregivers) from 14 special schools for IWSN, and professional caregivers of IWSN in 11 social institutions for people with disabilities in Jakarta, Indonesia. Other inclusion criteria for the respondents was age of at least 18 years old or older who can read and write in Indonesian language. Exclusion criteria was respondent who did not complete the survey or refused to participate in the study.

The researchers had a complete list of the institutions and schools clustered by five administrative cities in Jakarta. This list then subjected to cluster sampling technique with randomization using Microsoft Excel 2016 (Microsoft Corp, Redmont, Washington, USA). The researchers directly visited the selected institutions and schools in the list for data collection.

The number of sample size was calculated using a numerical descriptive sample size formula. In this formula, n represents the required sample size, the variable α is the significance level of 5% thus *zα* is established as 1.96, *s* corresponds to the standard deviation based on previous research by Wimardhani, et al. [21] which was 0.65, and *d* was the desired margin error of 0.1. Using this formula, the minimum sample number was calculated to be 162 respondents from each of the informal and professional caregivers.
$$n={\left(\frac{z\alpha \times s}{d}\right)}^{2}$$

### 2.3. Instrument

The Health Literacy in Dentistry-Indonesia Version (HeLD-ID) was utilized in this study to measure OHL. This instrument has been previously translated into Indonesian language by Rahardjo, et al. [22] from the original version developed by Jones et al. [23]. The instrument contained 29 questions, divided into 7 domains, namely (1) access, (2) understanding, (3) support, (4) utilization, (5) economic barriers, (6) receptivity, and (7) communication). The response to each question was provided based on a five-point likert scale, coded 0–4, which represents increased ease in undertaking the activity (0 = ‘unable to do’, to 4 = ‘without any difficulty’). The possible final score ranged from 0 to 116 (greater score indicating better oral health literacy). The final score subsequently divided by 29 to obtain respondent’s average OHL score for use in analysis.

### 2.4. Conduct of Study

Prior to data collection, the researchers requested permission from each head of the special school and social institution to conduct the onsite survey. Paper-based questionnaire was given directly to the respondents and it includes the information on research objectives and consent form. The respondents completed the questionnaire within ±10 min and returned the questionnaire directly to the researchers. Each respondent was only asked to participate once during the study between July and August 2022. Unreturned, blank, or incomplete questionnaire were considered as a refusal to participate and excluded from the data analysis.

### 2.5. Data Analysis

Collected data was analyzed using statistical software, IBM SPSS Statistics 22 (IBM Corp, Armonk, New York, NY, USA). Univariate analysis was conducted to describe the characteristics of caregivers. The total OHL score mean was tested with Kolmogorov-Smirnov test and showed abnormally distributed data. Thus, bivariate analysis was conducted using Kruskal-Wallis test to measure differences in oral health literacy scores between variables of IWSN caregivers’ characteristics. Next, to assess differences in OHL score between the two groups of informal and professional caregivers, another bivariate analysis using the Mann Whitney test was conducted. Analysis of results were considered significant when *p* < 0.05.

## 3. Results

Validity and reliability tests were performed before the main data collection on a subset of the caregiver respondents. The reliability test was performed using interclass correlation coefficient (ICC) test-retest to measure external consistency and questionnaire’s stability. Also, Cronbach’s alpha test was performed to analyze the questionnaire’s internal consistency. Every item in HeLD-ID had ICC score of 0.74–1 and Cronbach’s alpha score of 0.93 indicating that the questionnaire was reliable. Face validity test on respondents reported that none of the respondents had any problem understanding the questionnaire.

A total of 400 respondents participated in this study, consisting of 200 informal caregivers and 200 professional caregivers. Table 1 shows the demographic characteristics of respondents. The dominant age group was older in informal respondents and most of them were female. In both groups of respondents, they mostly graduated from high school, had additional non-formal education to care for IWSN, responsible for IWSN with single complexity, and their last dental visit was over a year ago. The informal respondents mostly cared for one IWSN, while professional respondents cared for more than three IWSNs.

Table 2 shows OHL score in informal caregivers. Among those who received a formal education, significant differences of median OHL scores were noted in a few domains namely economic barriers, access, communication, and utilization in relationship with their level of completed education. Additionally, significant differences in median OHL scores were noted among respondents in accordance with the number of IWSN that they manage. It was found that the median OHL scores were significantly lower among caregivers with more than one IWSN in the communication and utilization domains. The median OHL scores were also significantly different according to the caregivers’ last dental visit. Those whose last dental visits were more than one year ago had a significantly lower median OHL score in economic barriers, access, communication and utilization domains.

Among respondents of professional caregivers, there was a statistically significant difference in median OHL scores in several domains according to the age group, level of education, number of IWSN under their care, and their last dental visit (Table 3). It was found that the median scores differed across the age group in the access domain. Significant differences noted in the domains of receptivity, understanding, support, and economic barrier based on their level of completed education which understanding had the highest median OHL. Median scores in the access domain were also higher among respondents whose last dental visit was less than one year ago. The median scores for all seven domains were significantly higher among respondents who were handling more than 3 IWSN.

Table 4 depicts the differences in median OHL score between the two study groups across the seven domains, analyzed by Mann-Whitney test. There were significant differences noted between the groups in several domains namely in receptivity, understanding, and economic barrier (*p*-value < 0.05). In receptivity and understanding domains, the median value was higher in the informal caregivers. In contrast, the economic barrier median score was higher among professional caregivers. Finally, the total OHL score for professional caregivers was higher than the informal caregivers, although this difference was not statistically significant.

## 4. Discussion

This study was conducted in Indonesia, where awareness of oral health care for people with special needs has been increasing in recent years [24]. With diversity in culture and socio-economic status [24], Indonesian citizens, including IWSN and their caregivers, experience various barriers in achieving optimal personal and professional oral health care [25]. In order to strategically plan for improvement in oral health status of the nation, it is crucial that gaps in the current system are identified and addressed.

In this study, OHL was assessed as one of the integral factors towards establishing good oral health knowledge, attitudes and behavior [26]. A person with OHL would be able to make appropriate decisions by applying the information that they receive [12,27]. In other words, oral health inequalities could be minimized through acquisition of OHL, as people understand the importance and implications of taking appropriate action towards their oral health [12,27]. Furthermore, OHL influences oral health status since it can minimize inequality in oral health and improve the promotion of oral health information [12,27].

This study assessed the level of OHL of IWSN’s caregivers, who play an important role in various aspects of daily living of the person under their care. The HeLD-ID instrument was used to measure OHL in multiple facets, namely receptivity, understanding, support, economic barriers, access, communication, and utilization [22,23]. The majority of other OHL instruments only evaluate word recognition, reading comprehension, and computation which are considered as the main skills of OHL [27].

Initially, Jones and colleagues created the HeLD questionnaire by modifying the Health Literacy Measurement Scale (HeLMS) developed by Jordan et al. [23]. The HeLMS identified seven important abilities that a patient perceived to be crucial in seeking, understanding and utilising health information within the healthcare setting: knowing when to seek health information, knowing where to seek health information, verbal communication skills, assertiveness, literacy skills, capacity to process and retain information and application skills [28].

Oral diseases such as caries, periodontal problems, and malocclusions were prevalent in IWSN [29]. Moreover, IWSN heavily depended on their caregivers to make oral health decisions and perform oral health practices [15,30]. Studies have shown that there were lower oral health indices in children whose caregivers’ OHL were poor, suggesting a positive correlation between caregivers’ OHL with the oral health status of the individuals under their care, including IWSN [20,31].

It was found that the overall level of OHL among caregivers in this study was good, reflected by the median total score of above 3.0, respectively 3.14 (0.24–4) for informal caregivers and 3.21 (0–4) for professional caregivers. The scores were, however, lower in comparison with another study in the same country, which was conducted on caregivers of older adults living in public nursing homes [21]. Such a difference indicates the importance of training of caregivers in oral health care to encourage development of OHL. It was previously reported that initiatives to educate workers in Indonesian nursing homes about oral health care has been ongoing [32], which should similarly be implemented on caregivers of IWSN.

This study also found no significant difference in the overall score of OHL between informal and professional caregivers of IWSN attending special school, and those living in social institutions for people with disability. However, it was interesting to note that the OHL score for the economic barrier domain was significantly lower among respondents from special school, in comparison with their counterparts. In contrast, this group demonstrated significantly higher OHL in the receptivity and understanding domains. It is difficult to deduce the factors associated with such findings from this study. Further investigation is therefore recommended to explore reasons associated with low OHL in the specific domains among the respective respondents.

For both study groups, there were significant differences in OHL scores in certain domains according to the number of IWSN handled. It was noted that OHL scores were higher among caregivers with a greater number of IWSN handled; such findings were observed in two domains for respondents from special school and all domains for respondents from social institutions. A higher level OHL among those with a greater number of IWSN handled may be related to their experience as a caregiver. It was previously reported that parents with more than one IWSN under their care would develop better knowledge and confidence on how to manage their child, and improved understanding of the health care system [17].

Caregivers’ last dental visit was also associated with significant differences in OHL scores in certain domains within each study groups. It was noted that OHL scores were higher among caregivers whose last dental visit was less than one year ago; such findings were observed in four domains for respondents from special school and one domain for respondents from social institutions. Those who visit dentists regularly has been associated with better oral health knowledge and attitudes [33]. Therefore, it is deduced that higher OHL among those who demonstrate regular dental visit may be related to better knowledge and attitude towards oral health care. Developing positive oral health knowledge and attitudes among caregivers of IWSN is important to ensure that it is transformed onto better oral health maintenance of people under their care [6].

Improving communication skills related to oral health and access to oral healthcare services and information will not only improve the caregivers’ OHL but also the oral health condition of the individuals receiving their care [30]. Caregivers who often make dental visits and interact with oral health professionals form conceptual knowledge which increase their awareness towards oral health. This further creates a positive loop since high oral health awareness encourages caregivers to make more visits to oral care services [20]. Conversely, misinterpretation of instruction from oral health care professionals is associated with low OHL, which may result in serious errors. Moreover, individuals with low OHL are less likely to perform important preventive measures for maintaining oral health [34].

This study also showed that caregivers’ education correlated with OHL domains. Higher level of education has been shown to be correlated with better OHL in caregivers of children [35]. In young IWSN, caregivers with better knowledge of oral health translate into better attitude and practice towards oral health; knowledgeable caregivers will actively accompany the children doing tooth brushing and take them to get fluoride treatment [6].

Treatment for oral diseases creates financial burden to the family and healthcare system. Especially in IWSN, their oral diseases are more severe and frequent and may require complex management [36,37]. Meanwhile, poor oral health can also increase the risk of other health problems thus oral health care is an important issue to be addressed since it is related with individuals’ quality of life [37]. Indonesia has universal health coverage which can be optimized for IWSN who need oral health care.

This study investigated the level of OHL in the various domains within and between the two study groups. However, further studies to investigate the relationship between caregivers’ level of OHL with IWSN’s oral health status are recommended. Such studies would provide more information on the research matter, resulting in better understanding of the scenario and better strategies to remedy the issue. While such studies have been conducted elsewhere [20], similar studies must be conducted locally, should a specific and targeted approach be formulated and implemented.

The limitation and the weakness of this study was that the level of caregivers’ OHL cannot be confirmed with IWSN oral health status. At the time of the data collection, there was still a COVID-19 emergency restriction on direct oral examination for research so it could not be performed. Moreover, due to the survey was carried out using papers instead of electronic questionnaire, the researchers have limitation to obtain respondents from other cities. Thus the result may only be relevant for caregivers in Jakarta. However, since this study included a large number of both informal and professional caregivers, this became the strength of this study, especially when there was a lack of study on IWSN caregivers in Indonesia related to oral health.

Lastly, the researchers of this study proposed an intervention to IWSN caregivers such as in improving access to oral health care services. For example, dental schools can conduct oral health education and service programs in special schools which will be more convenient for both IWSN and their caregivers. This program in turn could benefit the students and dental educators by giving them clinical experiences so that they will be more comfortable in performing oral health care for IWSN. Since IWSN will continue to increase in the future, it is imperative that oral health care professionals are well trained and willing to perform special care dentistry. Lack of experience has been reported as the major reason why they deferred from doing special care dentistry in their daily practice [24].

## 5. Conclusions

This study was the first study that compared OHL between different groups of IWSN caregivers in Indonesia. This study found that the level of OHL among caregivers of IWSN in Indonesian special school and social institution is good. However, significant differences in OHL scores among caregivers between both study groups were noted in some domains, namely receptivity, understanding and the economic barrier. Within each study group, those with a greater number of IWSN handled and whose dental visit was less than one year ago demonstrated significantly higher OHL scores in some domains. Improving access to oral health care services may greatly improve IWSN oral health as well as caregivers’ OHL.

## Figures and Tables

**Table 1 dentistry-11-00221-t001:** Demographic characteristics of caregiver respondents.

Respondents’ Characteristics		Informal		Professional	
		n (n = 200)	%	n (n = 200)	%
Age	Mean Median Min–Max	42.88 42 17–65		32.57 30 19–67	
Age group	<20 20–39 40–59 ≥60	4 59 132 5	2 29.4 65.7 2.5	6 141 51 2	3 70.5 25.5 1
Gender	Male Female	16 184	8.0 92.0	100 100	50 50
Education	Elementary School Junior High School High School Diploma Bachelor/Profession	13 40 102 13 32	6.5 19.9 50.7 6.5 15.9	1 9 106 27 57	0.5 4.5 53 13.5 28.5
Additional non-formal education	Yes No	5 195	2.5 97.5	31 169	15.5 84.5
Complexity of disability	Single Complexity Multiple Complexity	175 25	87.1 12.4	115 85	57.2 42.3
Number of IWSN in care	1 person 2 persons 3 persons >3 persons	189 11	94 5.5	1 19 1 179	0.5 9.5 0.5 89.5
Caregiver last visit to the dentist	>1 year ago <1 year ago	62 138	30.8 69.0	73 127	36.5 63.5

**Table 2 dentistry-11-00221-t002:** Median OHL and demographic characteristics among informal caregivers.

Informal Caregivers		Median OHL Score (Min–Max)						
		Receptivity	Understanding	Support	Economic Barrier	Access	Communication	Utilization
Age group	<20	4 (2.6–4)	4 (3.33–4)	4 (3–4)	2.5 (1.33–4)	3.5 (1–4)	3.83 (3.5–4)	4 (3.4–4)
	20–39	3.4 (0.6–4)	4 (0–4)	3.33(0–4)	2.33 (0–4)	3.23 (0–4)	2.83 (0–4)	3 (0.6–4)
	40–59	3.6 (0–4)	4 (0–4)	3.33 (0–4)	2.67 (0–4)	3.5 (0–4)	3.16 (0–4)	3.2 (0–4)
	≥60	3.6 (2.6–4)	4 (2.67–4)	3.67 (2–4)	2 (0.67–4)	3.75 (2.25–4)	3.33 (1.17–3.83)	3.8 (2.8–4)
	*p*	0.476	0.819	0.289	0.957	0.639	0.283	0.166
Sex	Male	3.4 (2.2–4)	3.83 (2–4)	3.67 (1.33–4)	2.83 (0,67–4)	3 (1–4)	2.91 (1.17–4)	3 (2.4–4)
	Female	3.6 (0–4)	4 (0–4)	3.33 (0–4)	2.33 (0–4)	3.5 (0–4)	3.16 (0–4)	3.4 (0–4)
	*p*	0.809	0.768	0.537	0.280	0.650	0.551	0.989
Education	Elementary	3.8 (1–4)	4 (0.33–4)	3.33 (0–4)	2.67 (0–4)	3.5 (1–4)	3.33 (1–4)	3 (1–4)
	Junior High	3.4 (0–4)	4 (0–4)	3.33 (0–4)	1.67 (0–4)	3 (0–4)	2.33 (0–4)	3.2 (0–4)
	Senior High	3.6 (0–4)	4 (0–4)	3.33 (0–4)	2.33 (0–4)	3.12 (0–4)	3 (0–4)	3.1 (0–4)
	Vocational	4 (2.6–4)	4 (3.33–4)	4 (2.67–4)	3.33 (1.67–4)	4 (2–4)	3.83 (2.17–4)	4 (2.2–4)
	Bachelor or professional degree	3.8 (1.6–4)	4 (2–4)	3.67 (1.33–4)	3 (1–4)	4 (0–4)	3.5 (1.17–4)	3.7 (1.4–4)
	*p*	0.329	0.124	0.170	**0.001 ***	**0.015 ***	**0.007 ***	**0.024 ***
Additional non-formal education	Yes	3.8 (3.2–4)	4 (3.67–4)	3.33 (3–4)	3 (2–4)	4 (3–4)	4 (3.17–4)	4 (3.6–4)
	No	3.6 (0–4)	4 (0–4)	3.33 (0–4)	2.33 (0–4)	3.25 (0–4)	3.16 (0–4)	3.2 (0–4)
	*p*	0.366	0.212	0.637	0.123	0.063	**0.040 ***	**0.017 ***
Complexity of disability	Single	3.6 (0–4)	4 (0–4)	3.33 (0–4)	2.67 (0–4)	3.5 (0–4)	3.16 (0–4)	3.2 (0.6–4)
	Multiple	2.8 (0–4)	4 (1.33–4)	3.33 (0–4)	2.33 (0–4)	4 (0–4)	3.33 (0–4)	3.4 (0–4)
	*p*	0.109	0.304	0.337	0.85	0.864	0.795	0.786
Number of IWSN under care	1 person	3.6 (0–4)	4 (0–4)	3.33 (0–4)	2.33 (0–4)	3.5 (0–4)	3.16 (0–4)	3.4 (0–4)
	2 persons	3 (1.4–4)	3.33 (2–4)	3 (2–4)	2.67 (0–4)	2.75 (0–4)	2.67 (1.67–3.67)	2.4 (1.6–3.6)
	*p*	0.372	0.272	0.276	0.692	0.09	**0.049 ***	**0.026 ***
Last visit to dentist	Less than one year ago	3.6(0–4)	4 (0–4)	3.67 (0.33–4)	3 (0–4)	4 (1–4)	3.5 (1–4)	3.7 (0–4)
	More than one year ago	3.6 (0–4)	4 (0–4)	3.33 (0–4)	2.33 (0–4)	3 (0–4)	3 (0–4)	3 (0–4)
	*p*	0.603	0.955	0.115	**0.016 ***	**0.013 ***	**0.001 ***	**0.02 ***

Kruskal Wallis test, *: *p* < 0.05 significant diference.

**Table 3 dentistry-11-00221-t003:** Median OHL and demographic characteristics among professional caregivers.

Professional Caregivers		Median OHL Score (Min–Max)						
		Receptivity	Understanding	Support	Economic Barrier	Access	Communication	Utilization
Age group	<20	2.4 (2–3.8)	3.3 (2–4)	2.83 (1.3–4)	1.67 (0–3)	2.62 (2–4)	3.5 (1.5–3.83)	3.4 (1–4)
	20–39	3.2 (0–4)	3.67 (0–4)	3.67 (0–4)	3 (0–4)	3 (0–4)	3 (0–4)	3.3 (0–4)
	40–59	3.4 (0–4)	4 (0–4)	3.67 (0–4)	3 (0–4)	4 (0–4)	3.5 (0–4)	3.6 (0–4)
	≥60	3 (2–4)	3.33 (2.67–4)	3.5 (3–4)	2.83 (2.67–3)	3.12 (2.25–4)	3.41 (3–3.83)	3.8 (3.6–4)
	*p*	0.153	0.462	0.5	0.15	**0.028 ***	0.636	0.454
Sex	Male	3.4 (0.6–4)	3.67 (0.33–4)	3.67 (0.67–4)	3 (0–4)	3 (0.5–4)	3.16 (0.5–4)	3.4 (0.8–4)
	Female	3.2 (0–4)	3.67 (0–4)	3.67 (0–4)	3 (0–4)	3.12 (0–4)	3 (0–4)	3.2 (0–4)
	*p*	0.526	0.653	0.809	0.379	0.132	0.188	0.455
Education	Elementary	-	-	-	-	-	-	-
	Junior High	2.2 (1–4)	3 (1–4)	3 (1–4)	2 (0–4)	3 (0.25–4)	2.67 (0–4)	2.4 (0–4)
	Senior High	3.2 (0–4)	3.67 (0–4)	3.67 (0–4)	2.83 (0–4)	3 (0–4)	3 (0–4)	3.2 (0–4)
	Vocational	3 (1.8–4)	3.67 (2–4)	3 (1.33–4)	3 (0.67–4)	3 (0–4)	3.5 (0.83–4)	3.4 (0.6–4)
	Bachelor or professional degree	3.6 (0–4)	4 (0–4)	4 (0–4)	3.33 (0–4)	3.75 (0–4)	3.33 (0–4)	3.6 (0–4)
	*p*	**0.019 ***	**0.007 ***	**0.024 ***	**0.035 ***	0.314	0.176	0.263
Additional non-formal education	Yes	3.6 (1.8–4)	3.67 (1.33–4)	3.67 (1.33–4)	3 (0.33–4)	4 (0.75–4)	3.5 (1.67–4)	3.6 (0.8–4)
	No	3.2 (0–4)	3.67 (0–4)	3.67 (0–4)	3 (0–4)	3 (0–4)	3 (0–4)	3.2 (0–4)
	*p*	0.777	0.852	0.752	0.897	0.116	0.365	0.115
Complexity of disability	Single	3.4 (0–4)	3.67 (0–4)	3.33 (0–4)	3 (0–4)	3 (0–4)	3.16 (0–4)	3.2 (0–4)
	Multiple	3.2 (0–4)	3.67 (0–4)	3.67 (0–4)	3 (0–4)	3.25 (0–4)	3.16 (0–4)	3.4 (0–4)
	*p*	0.791	0.473	0.345	0.323	0.228	0.758	0.722
Number of IWSN under care	1–3 persons	2.4 (1–4)	2.33 (0.33–4)	2 (0.67–4)	1 (0–4)	2 (0–3.25)	1.83 (0–4)	2 (0–4)
	>3 persons	3.4 (0–4)	4 (0–4)	3.67 (0–4)	3 (0–4)	3.5 (0–4)	3.33 (0–4)	3.4 (0–4)
	*p*	**0.001 ***	**0.000 ***	**0.000 ***	**0.000 ***	**0.000 ***	**0.000 ***	**0.001 ***
Last visit to dentist	Less than one year ago	3.4 (1–4)	3.67 (0.33–4)	3.67 (0.67–4)	3 (0–4)	3.75 (0–4)	3.33 (0–4)	3.4 (0–4)
	More than one year ago	3.2 (0–4)	3.67 (0–4)	3.33(0–4)	3 (0–4)	3 (0–4)	3 (0–4)	3.2 (0–4)
	*p*	0.622	0.997	0.323	0.068	0.011 *	0.352	0.172

Kruskal Wallis test, *: *p* < 0.05 significant difference.

**Table 4 dentistry-11-00221-t004:** OHL score between informal and professional caregivers.

Domains	Median OHL (Min–Max)		*p*
	Informal Caregivers (n = 200)	Professional Caregivers (n = 200)	
Receptivity	3.6 (0–4)	3.3 (0–4)	**0.036 ***
Understanding	4 (0–4)	3.67 (0–4)	**0.030 ***
Support	3.33 (0–4)	3.67 (0–4	0.391
Economic Barrier	2.67 (0–4)	3 (0–4)	**0.022 ***
Access	3.5 (0–4)	3 (0–4)	0.976
Communication	3.00 (0–4)	3.16 (0–4)	0.728
Utilization	3.2 (0–4)	3.2 (0–4)	0.756
Average total OHL	3.14 (0.24–4)	3.21 (0–4)	0.495

Mann-Whitney test, *: *p* < 0.05 significant difference.

## Data Availability

The datasets used and analyzed during the current study are not publicly available due to subsequent undergoing study but are available from the corresponding author on reasonable request.

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
