# Peer review of "Analysis of Oral Health Literacy in Caregivers of Special Needs Individuals in Special Schools and Social Institutions in Jakarta"

_dentistry, 2023, doi:10.3390/dj11090221_

Round 1

Reviewer 1 Report

Thank you for allowing me to review this manuscript whose purpose was to analyze the oral health literacy in caregivers of individuals with spwcical needs in special schools (informal caregivers) and social institutions for people with disabilities (professional caregivers) in Jakarta, Indonesia.

Before accepting mauscript, it is necessary to make certain changes:

1. In the abstract - results - support them with numbers and p values.

2. In the introduction, introduce the readers to the subject in more detail, what is the oral health of people with special needs, how to improve oral health and the role of guardian parents in maintaining the oral health of people with special needs.

3. In the methods, explain in more detail the calculation of the sample size, as well as the inclusion and exclusion criteria for participation.

4. How was the distribution of the obtained data examined?

5. I think that all the results in Tables 2 and 3 could not be processed by the Kruskal Wallis test. Please put IQR next to min and max. It might be a coincidence that a regression analysis was chosen for the analysis.

6. The discussion insufficiently explains and compares the obtained results with other studies.

7. What is the strength of this study and its limitations.

8. The conclusion is a repetition of the results of the abstract and the results of that discussion, please conclude the mauscript.

9. References are not written in accordance with the instructions of the Journal.

Reviewer 2 Report

The submitted article titled "Analysis of Oral Health Literacy in Caregivers of Special Needs Individuals in Special Schools and Social Institutions in Jakarta" addresses a pertinent issue and is generally well-written. The chosen topic is both interesting and noteworthy, as it explores oral health literacy among caregivers of special needs individuals in specialized educational and social settings in Jakarta. The study's focus on a vulnerable population highlights the importance of understanding and addressing oral health literacy in this context.

The authors have effectively discussed the research methodology and findings. However, there is a minor concern regarding clarity in the statement found in rows 69-70 of the manuscript, where the authors mention the sample size calculation. It is recommended that the authors provide further explanation or elaboration on the numerical descriptive sample size formula employed. This will aid readers in understanding the basis of their sample size determination.

Another notable observation pertains to the presentation of results. The tables in the results section provide Kruskal-Wallis p-values, indicating the presence of differences. However, the lack of additional information on the specific groups where these differences exist necessitates further analysis. It is strongly suggested that the authors conduct post hoc tests to identify the specific group differences, which would enhance the interpretability of the findings.

Moreover, the discussion section would significantly benefit from the inclusion of a discussion of post hoc test results if conducted. Performing post hoc tests would offer additional insights into the variations observed among different groups, thereby adding depth to the interpretation of results and the overall discussion.

Furthermore, the manuscript lacks a dedicated section addressing the limitations of the study. It is essential for authors to candidly acknowledge and discuss the potential limitations of their research. This would enhance the credibility of the study and provide readers with a well-rounded understanding of the scope of the findings.

Lastly, there is a noted issue with the referencing style employed in the manuscript. The authors are kindly advised to carefully review the journal's Author Guidelines and make the necessary adjustments to ensure adherence to the prescribed referencing style.

Minor editing of English language required

Reviewer 3 Report

The aim of the paper was to evaluate the Oral health literacy ( OHL) in caregivers of special needs individuals in special schools and institutions in Indonesia. The paper is generally well written and easy to follow.

Introduction – A brief background of the study was clearly written.

Materials & Methods - The. Sample size of 400 is good to allow meaningful analysis of the data. While standard OHL questionnaire used previously was utilized, for the benefit of the readers, perhaps a brief description of the seven domains would be useful. It would be useful to give more information as to how the respondents were recruited, whether the caregivers were selected from certain schools/sectors or randomly recruited.

Results – The results were carefully organized and easy to follow. Perhaps some description of the results in the various domains could be given to allow better interpretation of the data.

Discussion - The salient points have been raised. Some variations in the responses from the different domains were found among the professional and informal caregivers, it would be useful to elaborate on the implications of the study arising from the results obtain.

Round 2

Reviewer 1 Report

Thanks the authors for accepting requested modifications.